# Prognostic value of neoadjuvant therapy for resectable and borderline resectable pancreatic cancer: A meta-analysis of randomized controlled trials

**Shangtong Liu, Hui Li\*, Yuhui Xue, Liang Yang**

Medical College, Jishou University, Jishou City, Xiangxi Tujia and Miao Autonomous Prefecture, Hunan Province, China

\* 461396419@qq.com

## Abstract

### Objective

To investigate the prognostic value of preoperative neoadjuvant therapy (NT) compared to upfront surgery (US) in patients with resectable and borderline resectable pancreatic cancer.

### Methods

PubMed, Embase, Web of Science were searched to collect randomized controlled trials on preoperative neoadjuvant therapy versus upfront surgery for resectable and borderline resectable pancreatic cancer before April 7, 2023, and data were extracted after screening according to inclusion and exclusion criteria, and HRs were obtained indirectly using enguage software; Stata 12.0 software was used for data analysis.

### Results

A total of 8 randomized controlled trials (RCTs) were included in this study, comprising a total of 1058 cases, including 503 cases in the NT group and 555 cases in the US group. Using an intention-to-treat population (ITT) analysis, the results showed that neoadjuvant treatment improved the R0 resection rate (RR 2.71, 95% CI 1.59–4.62; P = 0.000; $I^2$ = 46.20%) and overall survival (HR 0.66, 95% CI 0.54–0.82; P = 0.000; $I^2$ = 0.00%). In the subgroup of patients with resectable pancreatic cancer, the R0 resection rate in the NT group versus the US group (RR 1.14, 95% CI 0.93–1.39; P = 0.196; $I^2$ = 0.00%) and overall survival (HR 0.89, 95% CI 0.64–1.24; P = 0.489; $I^2$ = 0.00%) were not statistically significant.

### Conclusions

Preoperative neoadjuvant treatment is of prognostic value in patients with borderline resectable pancreatic cancer, as it increases the R0 resection rate and improves overall survival compared to upfront surgery.

**Data Availability Statement:** All relevant data are within the paper and its Supporting Information files.

**Funding:** The authors received no specific funding for this work.

**Competing interests:** The authors have declared that no competing interests exist.

## 1. Introduction

Recently, the incidence of pancreatic cancer has been increasing year by year due to the continuous improvement of examination means [1]. It has been reported that pancreatic cancer is the 7th most common malignant tumor in men and the 11th in women in China according to the statistics of China National Cancer Center in 2021 [2, 3]. The five-year survival rate for this disease is only 7.2%, highlighting the urgent need for improved treatment options [4]. In terms of imaging presentation, pancreatic cancer can be classified into three categories: resectable pancreatic cancer (RPC), borderline resectable pancreatic cancer (BRPC), and unresectable (locally advanced or metastatic) pancreatic cancer [5]. The criteria for resectable pancreatic cancer stipulate that the tumor should not invade the celiac trunk, superior mesenteric artery (SMA), and common hepatic artery (arterial requirements); in terms of venous requirements, the tumor should not invade the superior mesenteric vein (SMV) and portal vein (PV), or invade but not exceed 180 degrees with a regular vein profile. Alternatively, borderline resectable pancreatic cancer requires different criteria depending on the tumor location, as well as the affected arteries and veins [2]. The National Comprehensive Cancer Network (NCCN) guidelines recommend surgical resection as the primary treatment option for resectable pancreatic cancer, with neoadjuvant therapy suggested for high-risk factors; for borderline resectable pancreatic cancer, neoadjuvant therapy is the main treatment option [6]. Despite improvements in the diagnosis and treatment of pancreatic cancer, there has been a decline in five-year survival rates from 11.7% to 7.2% over the last two decades [4]. The conventional treatment for resectable pancreatic cancer is surgery and postoperative adjuvant chemotherapy, with patient selection based on imaging. However, prognosis still varies among patients receiving the same conventional treatment [7], even in those with pancreatic cancer without invasion and metastasis. Therefore, the search for new treatment options for resectable pancreatic cancer is urgent, in addition to actively exploring the aggressiveness indicators of resectability.

Neoadjuvant therapy is currently a research hotspot in oncology [8]. Previous randomized trial studies have confirmed its therapeutic value in esophageal and breast cancers [9, 10]. With several randomized controlled trials, the role of neoadjuvant therapy in borderline resectable pancreatic cancer is gradually being recognized [2]. The Korean multicenter phase II/III randomized controlled trial by Jang et al. demonstrated that neoadjuvant therapy improved the R0 resection rate (n = 14, 51.8% vs. n = 6, 26.1%, P = 0.004) and two-year survival rate (HR 1.97, 95% CI 1.07–3.62; P = 0.028) in patients with borderline resectable pancreatic cancer compared to the control group [11]. An observational study from three Australian centers also confirmed the feasibility and potential benefit of neoadjuvant therapy for patients with borderline resectable pancreatic cancer prior to surgical resection [12]. Based on these findings, the NCCN guidelines recommend neoadjuvant therapy as a priority for patients with borderline resectable pancreatic cancer [6]. Despite evidence supporting the benefits of neoadjuvant therapy for borderline resectable pancreatic cancer (BRPC), few high-quality prospective studies have been conducted to determine the optimal neoadjuvant regimen. Additionally, the role of neoadjuvant therapy in resectable pancreatic cancer (RPC) remains controversial. Nevertheless, the Italian randomized controlled trial by Reni et al. reported an R0 resection rate of 63% in the neoadjuvant group versus 37% in the surgery-first group, providing evidence for the effectiveness of neoadjuvant chemotherapy in resectable pancreas [13]. Conversely, the Dutch multicenter randomized controlled trial showed no significant difference in median survival between the neoadjuvant and surgery-first groups for intent-to-treat analysis in RPC (HR 0.96, 95% CI 0.64–1.44; P = 0.830) [14]. The Italian Casadei et al. randomized controlled clinical trial suggested no significant difference in R0 resection rates for intent-to-treat analysis in RPC between the neoadjuvant and surgical first groups (OR = 1.91, P = 0.489) [15].

In clinical practice, surgical resection remains the preferred treatment approach for resectable tumors [16]. However, the R0 resection rate for pancreatic cancer remains currently low, ranging from 15% to 20% [17]. The ability to achieve early R0 resection is crucial in determining the prognosis of resectable pancreatic cancer, as R0 resection is defined as the absence of residual tumor [18]. Despite the potential benefits of neoadjuvant therapy for RPC and BRPC, controversies persist regarding its R0 resection rate and prognostic value, with a lack of high-level evidence-based medical evidence. In this context, meta-analysis represents a common statistical tool for calculating the best estimates, overcoming sample differences between studies, and statistically analyzing controversial findings to draw clearer conclusions [19]. The aim of this study was thus to quantify the prognostic value of neoadjuvant therapy in RPC and BRPC.

## 2. Data and methods

### 2.1 Search strategy

Subject heading "resectable Pancreatic Neoplasms"、 entry terms "resectable Pancreatic Neoplasm", "resectable Pancreas Neoplasms", "resectable Pancreas Neoplasm", "resectable Cancer of Pancreas", "resectable Pancreas Cancers", "resectable Pancreas Cancer ", "resectable Pancreatic Cancer", "resectable Pancreatic Cancers", "resectable Pancreatic Cancer of the "resectable Cancer of the Pancreas", "Neoadjuvant Chemotherapy", and the free word "Neoadjuvant Chemotherapies", "Neoadjuvant Chemotherapy Treatment", "Neoadjuvant Chemotherapy Treatments", and "Neoadjuvant Chemotherapy Treatments". Neoadjuvant Chemotherapy Treatments" search terms in the title of Pubmed, Embase and Web of Science database, "cancer", "statistics" search The Pubmed database was searched by "cancer" and "statistics", and the CNKI and Wanfang databases were searched by the Chinese search terms of neoadjuvant treatment, neoadjuvant chemotherapy, neoadjuvant chemoradiotherapy, resectable pancreatic cancer, and pancreatic cancer; the end date of the search was April 7, 2023. All data and materials are available in Pubmed, Embase and Web of Science database.

### 2.2 Inclusion and exclusion criteria

**2.2.1 Inclusion criteria.** (1) patients with RPC and BRPC; (2) neoadjuvant therapy in the trial group (neoadjuvant therapy, NT group) and surgery in the control group (upfront surgery, US group); (3) providing outcome indicators such as R0 resection rate; (4) study type as RCT; (5) JADAD score of 4–7.

**2.2.2 Exclusion criteria.** (1) the study was conducted on patients with recurrent pancreatic cancer; (2) duplicate studies were conducted on patients from the same center; (3) the rate of missed visits exceeded 10% and the reasons for missing visits were not mentioned; (4) literature such as abstracts, letters, case reports, reviews or non-clinical studies were excluded.

### 2.3 Data extraction

The literature was meticulously evaluated in a blinded approach by two independent evaluators (Shan-Tong Liu and Liang Yang), and in the event of any discordant data acquisition, the evaluators consulted with each other to ensure consistency. The relevant information was recorded for each article and included the following entries: author, year, country, sample size, median (mean) age, use of neoadjuvant chemotherapy, use of adjuvant chemotherapy, follow-up time, fractionation, and treatment type. These entries are shown in Table 1. When the risk ratio (hazard ratio, HR) for overall survival (OS) was not explicitly presented in the literature,

**Table 1. Study characteristics.**

| author | year | Country | Samplesize (NT/US) | M(age) | | Neoadjuvant chemotherapy agent (NT) | Adjuvant chemotherapy agent (US) | Radiotherapy regimen | Follow up (mon) | Resectability status | Type | JADAD score | | | | |
|---|---|---|---|---|---|---|---|---|---|---|---|---|---|---|---|---|
| | | | | NT | US | | | | | | | ① | ② | ③ | ④ | ⑤ |
| Versteijne E | 2020 | Dutch | 246 (119/127) | 66 | 67 | gemcitabine | gemcitabine | 15 fractions of 2.4 Gy | 27 | R/BR | neoadjuvant CRT | 2 | 1 | 1 | 1 | 5 |
| Jang JY | 2018 | South Korea | 50(27/23) | 59.4 | 58.9 | gemcitabine | gemcitabine | 25 fractions of 1.8 Gy and 5 fractions of 1.8 Gy | 24 | BR | neoadjuvant CRT | 2 | 2 | 1 | 1 | 6 |
| Ghaneh P | 2023 | UK, German | 90(33/57) | 61 | 63.5 | B:gemcitabine, capecitabine: C: mFOLFIRINOX | A:5-FU/FA, gemcitabine | \ | 12.2 | BR | NAC | 2 | 2 | 1 | 1 | 6 |
| Casadei R | 2015 | Italy | 38(18/20) | 71.5 | 67.5 | gemcitabine | gemcitabine | 45 Gy and a boost of 9 Gy (tumor) | 40 | R | neoadjuvant CRT | 2 | 2 | 1 | 1 | 6 |
| Reni M | 2018 | Italy | 88(32/56) | 64 | 66.5 | C:PEXG | B:PEXG | \ | 55.4 | R | NAC | 1 | 1 | 1 | 1 | 4 |
| Golcher H | 2015 | Germany, Switzerland | 66(33/33) | 62.5 | 65.1 | Gemcitabine, cisplatin | gemcitabine | 1.8–55.8 Gy (tumor) or 50.4 Gy (regional lymph nodes) | 61 | R | neoadjuvant CRT | 2 | 1 | 1 | 1 | 5 |
| Seufferlein T | 2023 | Germany | 118(59/59) | 65 | 68 | nab-paclitaxel, gemcitabine | nab-paclitaxel, gemcitabine | \ | 60 | R | NAC | 2 | 1 | 2 | 1 | 6 |
| Motoi F | 2019 | Japan | 362 (182/180) | \ | \ | Gemcitabine, S-1 | S-1 | \ | 36 | R/BR | NAC | 2 | 2 | 1 | 1 | 6 |

NT, neoadjuvant therapy; US,upfront surgery; CRT, chemoradiotherapy; NAC, Neoadjuvant chemotherapy; BR, borderline resectable; R, resectable; mFOLFIRINOX, modified fluorouracil with folinic acid,irinotecan,oxaliplatin; PEXG, intravenous cisplatin 30 mg/m$^2$, epirubicin 30 mg/m$^2$, and gemcitabine 800 mg/m$^2$.

JADAD score: generation of random sequences (0–2 point), randomization concealment (0–2 point), blinding (0–2 point), withdrawal (0-1point)

the HR was calculated indirectly through enguage software by utilizing the provided survival curves in the original article.

### 2.4 Quality assessment

The quality of the literature included in the RCTs was evaluated by employing the JADAD score. This included the generation of random sequences (0–2 points), randomization conceal-ment (0–2 points), blinding (0–2 points), and withdrawal (0–1 point). The total JADAD score was 7, with a score of 1–3 being indicative of low quality and 4–7 being of high quality litera-ture; each literature was evaluated by two independent evaluators (Shan-Tong Liu and Yu-Hui Xue).

### 2.5 Statistic analysis

Meta-analysis was performed using Stata 12.0 software. The variables were the ratio of out-come indicators between patients in the NT and US groups, namely the relative risk of treat-ment (RR) or the risk ratio (HR). The I$^2$ test was used to measure the heterogeneity among the included studies, avoiding sample size biases. The fixed-effects model was utilized when the heterogeneity was insignificant (p>0.10 or I$^2$<50%) to generate the combined effect size value of RR or HR with a 95% confidence interval (95% CI). On the other hand, a random effects model was employed when the heterogeneity test p<0.10 or I$^2$>50%, signifying a significant heterogeneity. Subgroup analysis, sensitivity analysis, and meta-regression analysis were used to explore the sources of heterogeneity. Publication bias was assessed using Begg's test for quantitative detection, with a statistical significance of P<0.05 denoting its existence. ll manu-scripts must be in English, also the table and figure texts, otherwise we cannot publish your paper. Please keep a second copy of your manuscript in your office.

## 3. Results

### 3.1 Study selection

A comprehensive literature search was conducted on Pubmed, Embase, and Web of Science databases which yielded 2451 records. After removing duplicate records and non-RCTs, the title and abstract of 138 papers were screened. Subsequently, 14 papers underwent full-text analysis, leading to the inclusion of 8 RCT studies involving 1058 patients in the meta-analysis [11, 13–15, 20–23] (Fig 1).

### 3.2 Study characteristics

The table of characteristics of the included literature is shown in Table 1. Notably, 4 out of the 8 studies enlisted resectable pancreatic cancer [13, 15, 20, 22], while 2 studies included borderline resectable pancreatic cancer [11, 21]; the remaining two studies incorporated both resectable and borderline resectable pancreatic cancer [14, 23]. Of the 503 patients assigned to neoadjuvant therapy, 197 received neoadjuvant chemoradiotherapy (CRT) and 306 received neoadjuvant chemotherapy.

### 3.3 R0 resection rate

Seven randomized controlled trials (RCTs) reported R0 resection rates [11, 13–15, 20–22]. Neoadjuvant treatment significantly improved the R0 resection rate in comparison with controls (risk ratio [RR] 1.36, 95% confidence interval [CI] 1.12–1.64; P = 0.002; $I^2$ = 0.00%) (S1 Fig). Notably, a heterogeneity analysis showed no statistically significant heterogeneity across studies ($I^2$ = 0.00%). Therefore, a fixed-effects model was used for the combined analysis. The analysis revealed a positive effect of neoadjuvant therapy on R0 resection rates in patients with resectable and borderline resectable pancreatic cancer.

### 3.4 OS

Three randomized controlled trials (RCTs) reported hazard ratios (HRs) for overall survival (OS) [11, 14, 23], while two studies obtained HRs using enguage software to analyze the

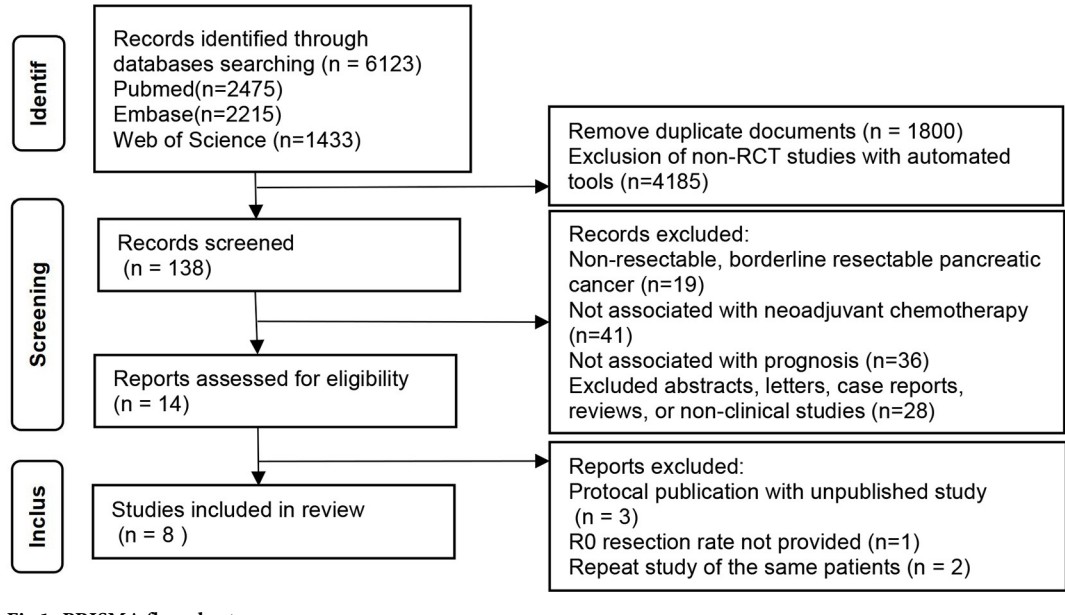

**Fig 1. PRISMA flow chart.**

survival curves provided in the original article [13, 22]. As there was no statistical heterogeneity among the studies ($I^2$ = 0.00%), a fixed-effects model was used for the combined analysis. The analysis revealed a positive effect of neoadjuvant therapy on overall survival in patients with resectable and borderline resectable pancreatic cancer (HR 0.72, 95% confidence interval [CI] 0.61–0.87; P = 0.000; $I^2$ = 0.00%) (S2 Fig).

### 3.5 Subgroup analyses

**3.5.1 Resectability status.** Subgroup analysis of patients with borderline resectable pancreatic cancer showed a statistically significant effect of neoadjuvant therapy on improving R0 resection rate (RR 2.71, 95% CI 1.59–4.62; P = 0.000) and overall survival (HR 0.66, 95% CI 0.54–0.82; P = 0.000; $I^2$ = 0.00%). However, in patients with resectable pancreatic cancer, no significant difference between the two groups in the R0 resection rate (HR 0.89, 95% CI 0.64–1.24; P = 0.489; $I^2$ = 0.00%) and the overall survival (RR 1.14, 95% CI 0.93–1.39; P = 0.196; $I^2$ = 0.00%) was observed (Fig 2A and 2B).

**3.5.2 Chemotherapy and chemoradiotherapy.** In the analysis of different neoadjuvant treatment types, neoadjuvant chemoradiotherapy improved the R0 resection rate (RR 1.43, 95% CI 1.11–1.84; P = 0.006; $I^2$ = 0.00%)) (S3 Fig) and the overall survival (RR 0.74, 95% CI 0.58–0.95; P = 0.018; $I^2$ = 0.00%)) (S4 Fig) of patients with resectable and borderline resectable pancreatic cancer compared to upfront surgery. Additionally, subgroup analysis of neoadjuvant chemotherapy showed that neoadjuvant chemotherapy was associated with longer overall survival (RR 0.70, 95% CI 0.54–0.91; P = 0.008; $I^2$ = 0.00%) compared with upfront surgery, whereas the two groups was found to be comparable, as illustrated in Table 2.

### 3.6 Sensitivity analysis and publication bias

We performed sensitivity analyses using Stata 12.0 by excluding each study, with R0 resection rate as the primary indicator and OS combined effect size stability as the secondary indicator. Publication bias was evaluated using Begg's funnel plot (Pr>|z| = 0.764) and Egger's linear regression (P>|t| = 0.480) (P<0.05 indicating publication bias), and no significant publication bias was identified (S5–S8 Figs).

## 4. Discussion

Numerous studies have investigated the prognostic value of neoadjuvant therapy in resectable and borderline resectable pancreatic cancer (BRPC); however, inconsistency persists among their results and opinions regarding the role of neoadjuvant therapy in treating resectable pancreatic cancer (RPC) remains controversial [24]. Therefore, we aim to investigate the prognostic value of neoadjuvant therapy in both RPC and BRPC via meta-analysis to obtain precise estimations.

This study comprised eight papers [11, 13–15, 20–23] that investigated the prognostic value of neoadjuvant therapy or upfront surgery in 1058 patients with resectable pancreatic cancer and borderline resectable pancreatic cancer. The results revealed that patients with borderline resectable pancreatic cancer (BRPC) who received neoadjuvant therapy had a significant improvement in overall survival (HR 0.66, 95% CI 0.54–0.82; P = 0.000; $I^2$ = 0.00%). This finding could be attributed to the potential of neoadjuvant therapy to reduce the stage of borderline resectable disease [25], thereby increasing the possibility of achieving an R0 resection during surgical resection.

Farren et al. analyzed the tumor microenvironment from another perspective and found that both neoadjuvant therapy and chemoradiotherapy could alter the pancreatic cancer microenvironment. Tumors receiving FOLFIRINOX had more intersecting immune gene

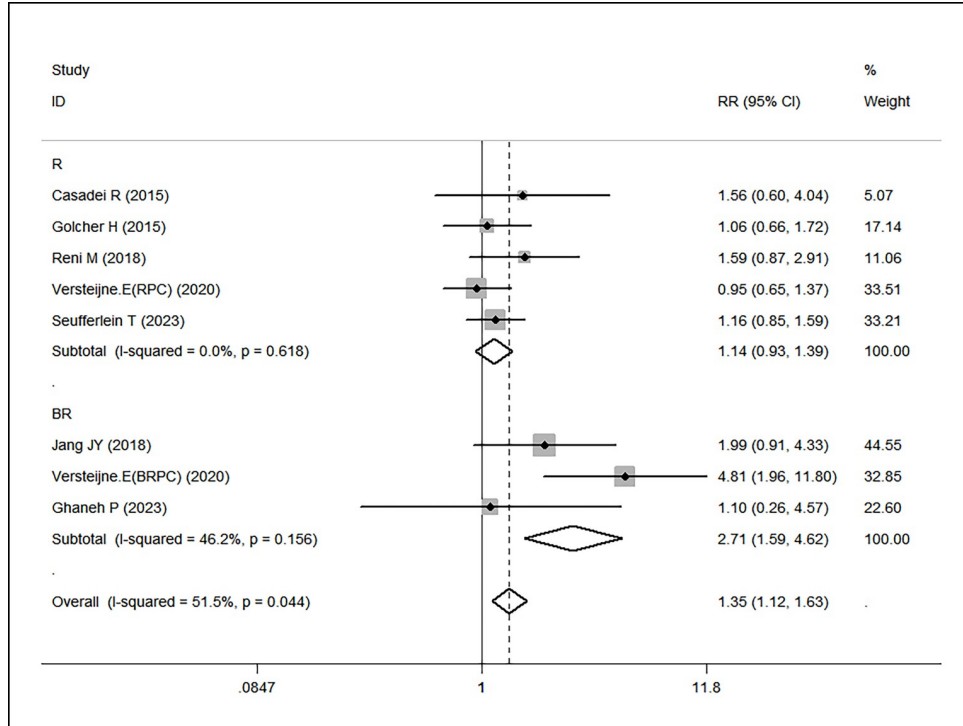

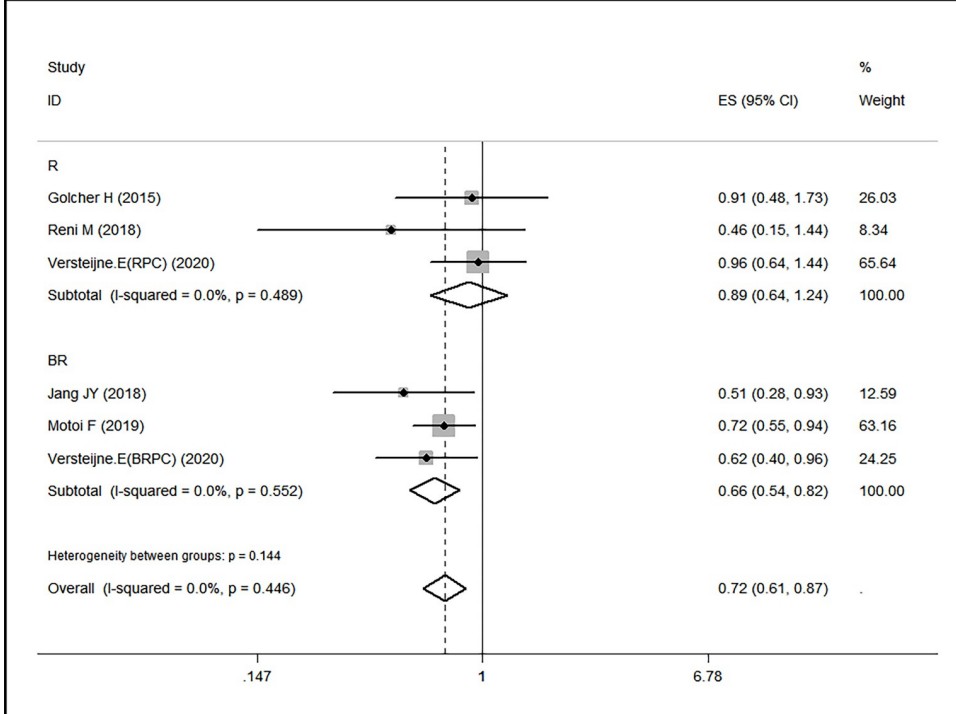

**Fig 2. R0 resection rate with subgroups for resectability status, b. a.** Overall survival with subgroups for resectability status.

**Table 2. Summary of the meta-analysis results.**

| Name | R0 resection rate | | | Heterogeneity | Overall survival | | | Heterogeneity |
|---|---|---|---|---|---|---|---|---|
| | N | RR (95%CI) | P | $I^2$ | N | HR (95%CI) | P | $I^2$ |
| | 7 | 1.36 (1.12,1.64) | 0.002 | 0.00% | 5 | 0.72 (0.61,0.87) | 0.000 | 0.00% |
| Subgroup 1: resectability status | | | | | | | | |
| R | 5 | 1.14 (0.93,1.39) | 0.196 | 0.00% | 3 | 0.89 (0.64,1.24) | 0.489 | 0.00% |
| BR | 3 | 2.71 (1.59,4.62) | 0.000 | 46.20% | 3 | 0.66 (0.54,0.82) | 0.000 | 0.00% |
| Subgroup 2: type | | | | | | | | |
| neoadjuvant CRT | 4 | 1.43 (1.11,1.84) | 0.006 | 0.00% | 3 | 0.74 (0.58,0.95) | 0.018 | 0.00% |
| NAC | 3 | 1.26 (0.95,1.66) | 0.111 | 0.00% | 2 | 0.70 (0.54,0.91) | 0.008 | 0.00% |

NAC, neoadjuvant chemotherapy; CRT, chemoradiotherapy; R, resectable resectable pancreatic cancer; BR, borderline resectable pancreatic cancer

networks, including interferon-related transcripts, cytokine and NF-κB-related transcripts, and complement cascade-related transcripts, as well as higher levels of CD8a, CD8b, co-stimulatory molecules, and molecules related to immune checkpoints [26].

Conversely, no significant difference was found in the R0 resection rate (RR 1.14, 95% CI 0.93–1.39; P = 0.196; $I^2$ = 0.00%) and overall survival (HR 0.89, 95% CI 0.64–1.24; P = 0.489; $I^2$ = 0.00%) of patients with resectable pancreatic cancer (RPC) who underwent neoadjuvant therapy compared to upfront surgery. This may be due to the significant delay in treatment caused by the lengthy period of disease that must pass before receiving neoadjuvant therapy. Pathological verification is necessary before adjuvant therapy, and an original diagnosis of RPC may progress to BRPC or a locally progressive form, which could in turn delay the timing of treatment [27].

Upon reviewing the literature from the included RCT studies, we identified inconsistent conclusions among studies regarding whether neoadjuvant therapy improves R0 resection rates. Therefore, we conducted an analysis of R0 resection rates across studies and speculated that the discrepancy in findings may be related to the method of analysis of R0 resection rates used by Versteijne et al. which followed a Per-Protocol (PP) approach [14]. In contrast, Casadei et al. calculated R0 resection rates using an intention-to-treat (ITT) population [15]. The PP analysis dataset is the compliance protocol set, which may exaggerate the effect of neoadjuvant therapy, while the ITT analysis considers the full analysis set and ensures comparability between groups. As a result, we chose to use ITT analysis in this paper.

In the subgroup analysis of neoadjuvant treatment types, neoadjuvant chemoradiotherapy improved the R0 resection rate in patients with resectable pancreatic cancer (RPC) and borderline resectable pancreatic cancer (BRPC) compared with the upfront surgery group (risk ratio (RR) 1.43, 95% confidence interval (CI) 1.11–1.84; P = 0.006; $I^2$ = 0.00%). Both neoadjuvant chemotherapy (hazard ratio (HR) 0.70, 95%CI 0.54–0.91; P = 0.008; $I^2$ = 0.00%) and neoadjuvant chemoradiotherapy (HR 0.74, 95%CI 0.58–0.95; P = 0.018; $I^2$ = 0.00%) were associated with improved overall survival compared with upfront surgery. In a meta-analysis conducted by Hajibandeh et al., the prognostic value of neoadjuvant chemoradiotherapy versus upfront surgery for resectable and borderline resectable pancreatic cancer was evaluated. The study demonstrated that neoadjuvant chemoradiotherapy led to a higher R0 resection rate (RR 1.55, p = 0.004) and longer median survival (MD: 3.75 years, p = 0.009) [28]. Although it is evident that both neoadjuvant chemotherapy and chemoradiotherapy have prognostic value for RPC and BRPC patients, there remains no high-level evidence-based evidence pointing to which is the best option. Future neoadjuvant treatment research for pancreatic cancer should aim to address this gap in knowledge.

There is a lack of high-quality evidence-based research for identifying optimal neoadjuvant chemotherapy regimens [29]. The Chinese guidelines for comprehensive diagnosis and treatment of pancreatic cancer recommend several agents for neoadjuvant chemotherapy, including FOLFIRINOX, gemcitabine (GEM) + albumin-bound paclitaxel, GEM + S-1 or GEM alone [30]. Nonetheless, among the eight studies included in the current meta-analysis, neoadjuvant chemotherapy regimens were primarily based on gemcitabine, highlighting the absence of a standardized neoadjuvant chemotherapy regimen and continued controversy surrounding the best chemotherapeutic agent for this indication. In a study by Ye et al., which examined the effect of neoadjuvant chemotherapy on overall survival (OS) in resectable pancreatic cancer, gemcitabine-based regimens (e.g., gemcitabine plus nab-paclitaxel, gemcitabine plus 5-FU) were found to be more effective than 5-FU-based regimens (e.g., full-dose or modified FOLFIRINOX and S-1 plus nab-paclitaxel) [31]. Conversely, no significant difference in overall survival was observed between the FOLFIRINOX and GEM/nab-PTX groups in a phase II clinical trial study conducted by Yamaguchi et al. [32]. A clinical trial study from the United States found that RPC patients who received preoperative selicrelumab exhibited tumor fibrosis and greater T-cell enrichment, suggesting that neoadjuvant chemotherapy with the CD40 monoclonal antibody selicrelumab altered the tumor microenvironment in resectable pancreatic cancer [33]. Choosing appropriate chemotherapeutic agents can increase the efficacy of neoadjuvant therapy, and the selection of neoadjuvant chemotherapy regimens remains a key research topic. More RCT studies are needed to provide a solid foundation for clinical guidance on the selection and use of neoadjuvant chemotherapeutic agents for pancreatic cancer treatment.

There are several limitations to this study:

Firstly, the basic characteristics of patients in the last included study differ in terms of ethnicity, resectability staging criteria, type and dose of chemotherapy drugs, mode and dose of chemoradiotherapy, and surgery, which may have impacted the results.

Secondly, the sample size of the included studies is small, thereby reducing the statistical power of the analysis.

Thirdly, the resectability staging criteria for pancreatic cancer in each study were not consistent, highlighting the need for future studies to adopt consistent criteria.

Fourthly, publication bias was assessed using the Egger test and Begg test, which may have limited efficacy when the number of studies included in the meta-analysis is less than 10; therefore, these tests may not have detected publication bias in this study.

Lastly, the literature search was limited to English language papers, possibly introducing language bias.

## 5. Conclusion

To summarize, the meta-analysis of these 8 articles confirms that neoadjuvant treatment of resectable pancreatic cancer is of prognostic value, increasing the R0 resection rate and improving overall survival compared with upfront surgery. However, uncertainty remains whether neoadjuvant therapy can improve the overall survival and R0 resection rate for patients with resectable pancreatic cancer. Future studies should explore whether neoadjuvant chemoradiotherapy is superior to neoadjuvant chemotherapy in the prognosis of borderline resectable pancreatic cancer (BRPC) patients, as well as the choice of neoadjuvant chemotherapy regimen, such as whether the FOLFIRINOX chemotherapy regimen is superior to gemcitabine-based chemotherapy regimens. Furthermore, more randomized controlled trials should be conducted in the future to obtain as many prognostic indicators as possible, providing more useful information and a scientific basis for the treatment of resectable pancreatic cancer and BRPC.

## Supporting information

**S1 File. Statistical data.**
(XLSX)

**S1 Fig. R0 resection rate.**
(TIF)

**S2 Fig. Overall survival.**
(TIF)

**S3 Fig. R0 resection rate with subgroups for chemotherapy and chemoradiotherapy.**
(TIF)

**S4 Fig. Overall survival with subgroups for chemotherapy and chemoradiotherapy.**
(TIF)

**S5 Fig. Sensitivity analysis.**
(TIF)

**S6 Fig. Begg's test.**
(TIF)

**S7 Fig. Egger's test.**
(TIF)

**S8 Fig. Funnel plot.**
(TIF)

## Acknowledgments

### Registration

The protocol of this study was registered in the International Prospective Register of Systematic Reviews (PROSPERO) with the registration number CRD42023413885.

## Author Contributions

**Conceptualization:** Shangtong Liu, Hui Li, Yuhui Xue, Liang Yang.

**Data curation:** Shangtong Liu, Yuhui Xue, Liang Yang.

**Formal analysis:** Shangtong Liu.

**Investigation:** Shangtong Liu, Hui Li, Yuhui Xue, Liang Yang.

**Methodology:** Shangtong Liu, Hui Li, Yuhui Xue.

**Resources:** Shangtong Liu, Yuhui Xue.

**Software:** Shangtong Liu.

**Supervision:** Shangtong Liu, Hui Li, Liang Yang.

**Validation:** Shangtong Liu, Yuhui Xue, Liang Yang.

**Visualization:** Shangtong Liu.

**Writing – original draft:** Shangtong Liu, Yuhui Xue.

**Writing – review & editing:** Shangtong Liu, Hui Li.

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
