## [Decision Letter · Decision Letter 0]

5 Jun 2023

PONE-D-23-13122Prognostic value of neoadjuvant therapy for resectable and borderline resectable pancreatic cancer: a meta-analysis of randomized controlled trialsPLOS ONE

Dear Dr. Liu,

Thank you for submitting your manuscript to PLOS ONE. After careful consideration, we feel that it has merit but does not fully meet PLOS ONE’s publication criteria as it currently stands. Therefore, we invite you to submit a revised version of the manuscript that addresses the points raised during the review process.

We look forward to receiving your revised manuscript.

Kind regards,

Andrea D’Aviero

Academic Editor

PLOS ONE

Reviewers' comments:

Reviewer's Responses to Questions

**Comments to the Author**

1. Is the manuscript technically sound, and do the data support the conclusions?

Reviewer #1: Yes

Reviewer #2: Partly

2. Has the statistical analysis been performed appropriately and rigorously? 

Reviewer #1: Yes

Reviewer #2: Yes

3. Have the authors made all data underlying the findings in their manuscript fully available?

Reviewer #1: Yes

Reviewer #2: Yes

4. Is the manuscript presented in an intelligible fashion and written in standard English?

Reviewer #1: Yes

Reviewer #2: No

5. Review Comments to the Author

Reviewer #1: The authors propose an interesting systematic review with meta-analysis of randomized clinical trials designed to evaluate the role of neoadjuvant treatment of resectable or borderline resectable pancreatic adenocarcinoma.

The English is correctly and adequately used, the methodology is indeed rigorously systematic, the results are well illustrated and justify the conclusions and the discussion, complete with evaluation of the study's limitations, in relation to the main reference literature.

My only suggestion to the authors: different types of neoadjuvant therapies are considered in aggregate in the results, opening up the range of which modalities and which treatment schemes only under discussion. In my opinion the paper would benefit from a short description in a dedicated paragraph within the results of the different types of neoadjuvant treatment regimens considered in the different trials already in the results to include, if available, data on radiotherapy schedules (dose, volumes, fractions).

Reviewer #2: The authors of this meta-analysis discuss the role of neoadjuvant RT in the context of resectable and borderline resectable pancreatic cancer.

Introduction.

- When the authors comment on the variability of prognosis in the context of resectable patients receiving the same treatment, I would add the reference.

- In this context, the authors talk about the need to look for biological markers of resectability, but I would ask the authors to discuss the need to look for markers of aggressiveness and thus different prognosis.

- I would remove sentences and references regarding the role of neoadjuvant treatments in oesophagus and breast from the introduction.

Data and methods

- No specific comment

Outcomes

- Is there a benefit over R0 in resectable patients with RT or chemoradiotherapy?

Discussion

- The authors only mention R0 and OS. Why these outcomes and not local control and distant metastasis free survival? These are also indicators that may differ depending on neoadjuvant therapy and have an impact on OS and quality of life.

- I would like to comment on the mechanism of alteration of the pancreatic tumour microenvironment described in Farren's study.

- Did the authors evaluate the effect of adjuvant therapies?

- What is the dose and fractionation of radiotherapy?

- Add an explanation of the acronyms 'R' and 'BR' in the legend of Table 2.

6. PLOS authors have the option to publish the peer review history of their article (what does this mean?). If published, this will include your full peer review and any attached files.

Reviewer #1: No

Reviewer #2: No

---

## [Author Response · Author response to Decision Letter 0]

14 Jun 2023

Dear Editor and Reviewers,

Thank you for taking the time to review our manuscript. We appreciate your insightful comments and have prepared this sincere and friendly response to each of the points raised:

In response to PLOS ONE Editor:

Thank you very much for providing us with the required revisions. We have made the necessary modifications in accordance with the guidelines provided by the PLOS ONE journal. Additionally, we have included a statement concerning funding in our cover letter, affirming that we have not received any financial assistance.

Furthermore, we have updated the ORCiD iD as requested by the editor. We apologize for the oversight on our part and appreciate your patience and guidance in this matter.

Thank you again for your continued support and assistance.

In response to Reviewer #1's suggestion: 

Thank you for your assistance in refining our manuscript and for your positive feedback. We greatly appreciate your suggestions to enhance the quality of our research. We have carefully considered your comments and have made the following modifications:

Results Section: We have included a dedicated paragraph analyzing different subgroups of resectability and neoadjuvant therapy. This addition provides a more comprehensive analysis of our findings and strengthens the organization and logical flow of our manuscript.

Table 1: We have added a new column to Table 1 that briefly describes the radiation therapy plans used in the neoadjuvant chemotherapy regimen. This includes relevant information such as dose, fractionation, and any available radiotherapy data. This addition provides important details for readers to better understand the treatment protocols utilized in our study.

We sincerely appreciate your thorough review of our research and your constructive feedback, as it has significantly contributed to the improvement of our work. If you have any additional suggestions or questions, we are more than happy to address them. Please feel free to contact us.

In response to Reviewer #2's comments:

1. Comment: When the authors comment on the variability of prognosis in the context of resectable patients receiving the same treatment, I would add the reference.

Response: We sincerely apologize for any concerns caused by the lack of rigor in our research. We appreciate your valuable feedback and assure you that we are fully committed to enhancing the quality of our work. We acknowledge the importance of providing appropriate references to support our statements.

Considering your suggestion, we have carefully reviewed our manuscript and have made the necessary revisions. We have included additional references that support our comment on the variability of prognosis in resectable patients receiving the same treatment. These additional references strengthen our argument and add value to the study by providing relevant evidence to support our claims.

2. Comment: In this context, the authors talk about the need to look for biological markers of resectability, but I would ask the authors to discuss the need to look for markers of aggressiveness and thus different prognosis.

Response: We appreciate your valuable input and sincerely apologize for any confusion caused by our previous statement. We have carefully revised the manuscript to address your comment and emphasize the importance of investigating markers of aggressiveness in addition to markers of resectability.

After thoroughly reviewing the literature on resectable pancreatic cancer, we acknowledge that the prognosis of resectable pancreatic cancer can vary even when patients receive the same treatment. This highlights the need for personalized treatment approaches to optimize outcomes. In line with your suggestion, we have expanded our discussion to include the necessity of identifying markers of aggressiveness that can aid in accurately assessing the prognosis of resectable pancreatic cancer.

Currently, the determination of resectability in pancreatic cancer is primarily based on imaging criteria, which may not always accurately assess the extent of invasion. Therefore, the identification of markers associated with aggressiveness would be crucial in improving the accuracy of prognostic assessment in resectable pancreatic cancer cases.

Furthermore, we would like to highlight that new treatment options for pancreatic cancer are emerging, offering hope and promise for patients with this devastating disease. The discovery and validation of markers of aggressiveness can not only aid in prognosis assessment but also guide the selection of appropriate therapeutic strategies.

3. Comment: I would remove sentences and references regarding the role of neoadjuvant treatments in oesophagus and breast from the introduction.

Response: Thank you for your valuable feedback on our manuscript. We sincerely appreciate the time and effort you have put into reviewing our article and providing constructive comments.

Regarding the inclusion of references to neoadjuvant treatments for esophageal and breast cancer in the introduction, we understand your concern. We apologize if this inclusion has affected the focus of our manuscript. We appreciate your suggestion to remove this section, and we acknowledge that it may help streamline the content and maintain a more focused discussion.

After careful consideration, we have decided to follow your recommendation and remove the sentences regarding the role of neoadjuvant treatments in esophageal and breast cancer from the introduction. By doing so, we aim to ensure that the manuscript maintains a clear and cohesive focus on neoadjuvant therapy specifically in the context of pancreatic cancer.

4. Comment: Is there a benefit over R0 in respectable patients with RT or chemoradiotherapy?

Response: Thank you for your insightful comment regarding the potential benefit of achieving an R0 resection in resectable patients receiving radiotherapy or chemoradiotherapy. We apologize for any confusion caused by our imprecise terminology in the manuscript.

In our study, we focused specifically on investigating the impact of neoadjuvant chemotherapy and neoadjuvant chemoradiotherapy on the prognosis of patients with resectable or borderline resectable pancreatic carcinoma. We did not include neoadjuvant radiotherapy as a separate treatment modality in our analysis.

To address your concern and provide clarity, we have carefully reviewed our manuscript and made appropriate corrections. We have revised the terminology used to accurately reflect the treatment options studied in our randomized controlled trial. Specifically, we have used the term "neoadjuvant chemoradiotherapy" instead of "radiotherapy" to better align with our research focus.

5. Comment: The authors only mention R0 and OS. Why these outcomes and not local control and distant metastasis free survival? These are also indicators that may differ depending on neoadjuvant therapy and have an impact on OS and quality of life.

Response: Thank you for your insightful comments on our manuscript. We appreciate your observation regarding the inclusion of additional outcome measures such as local control and distant metastasis-free survival in our analysis. These outcomes are indeed important indicators that can impact overall survival and quality of life in patients receiving neoadjuvant therapy.

We agree that local control and distant metastasis-free survival are valuable endpoints to assess the efficacy of neoadjuvant therapy. However, during our data extraction process, we found that the studies included in our analysis reported a wide range of outcomes, and there was a lack of consistency in the reporting of these specific measures. The majority of studies consistently reported R0 resection rate and overall survival, making them the most suitable outcome measures for our analysis. Unfortunately, the limited availability and variability of data on local control and distant metastasis-free survival prevented us from including them as additional outcomes in our study.

We recognize the significance of local control and distant metastasis-free survival as important prognostic indicators and their impact on overall survival and quality of life. The limitations of our study, including the choice of outcome measures, have been duly noted. We appreciate your suggestion for future research to explore these outcomes more comprehensively in the context of neoadjuvant therapy. Moving forward, we will consider these valuable insights as we continue to refine and expand our research in this field. 

6. Comment: I would like to comment on the mechanism of alteration of the pancreatic tumour microenvironment described in Farren's study.

Response: Thank you for your comment regarding the mechanism of alteration in the pancreatic tumor microenvironment described in Farren's study. We appreciate your interest in understanding the underlying mechanisms behind the changes observed in the tumor microenvironment.

In Farren's study, they employed various approaches to investigate the impact of neoadjuvant chemotherapy on the tumor microenvironment. They assessed changes in signaling pathways, immune checkpoint expression, and Pan-immune cell markers to gain insights into the mechanisms involved.

The results of Farren's study revealed several noteworthy findings. Firstly, neoadjuvant therapy led to alterations in the expression levels of immune-related proteins within the tumor microenvironment. Notably, higher levels of granulysin B and PD-L1 were observed in the enriched tumor areas following neoadjuvant therapy.

Furthermore, significant differences were observed in the expression of B cell markers CD19 and CD20, as well as the monocyte marker CD14, between the neoadjuvant therapy group and the upfront surgery group. These differences suggest that neoadjuvant therapy impacts the composition and activity of specific immune cell populations within the tumor microenvironment.

In addition to immune-related changes, Farren's study also identified differences in signaling pathways involved in regulating migration, survival, and other cellular activities. These pathways included pSTAT3, PTEN, and BCL-2, which showed significant differences between the neoadjuvant therapy group and the upfront surgery group.

Overall, Farren's study provides valuable insights into the mechanisms by which neoadjuvant chemotherapy can influence the pancreatic tumor microenvironment. These findings shed light on the complex interplay between the tumor and its surrounding microenvironment, highlighting the potential impact of neoadjuvant therapy on immune response and signaling pathways.

We hope this response has addressed your questions regarding the mechanisms described in Farren's study. Should you have any further inquiries or require additional clarification, please feel free to contact us.

7. Comment: Did the authors evaluate the effect of adjuvant therapies?

Response: Thank you for your comment regarding the evaluation of adjuvant therapies in our study. We appreciate your interest in understanding the impact of adjuvant therapies on the outcomes of patients with resectable and borderline resectable pancreatic cancer.

In our study, we focused primarily on investigating the efficacy of neoadjuvant therapy compared to the conventional approach of upfront surgery followed by adjuvant therapy. We included randomized controlled trials that assigned patients to either the neoadjuvant therapy group or the upfront surgery followed by adjuvant therapy group.

The aim of our analysis was to compare the outcomes between these two treatment approaches, specifically evaluating the impact of neoadjuvant therapy on R0 resection rates and overall survival. We found that neoadjuvant therapy resulted in higher R0 resection rates and improved overall survival compared to upfront surgery alone.

However, it is important to note that our analysis did not directly assess the effect of adjuvant therapies alone, as we focused on the comparison between neoadjuvant therapy and upfront surgery followed by adjuvant therapy. The effectiveness of adjuvant therapy in our study was evaluated within the context of the overall treatment approach.

We acknowledge that the evaluation of adjuvant therapies alone is an important aspect that merits further investigation. Future studies specifically designed to assess the effectiveness of adjuvant therapies in pancreatic cancer patients could provide valuable insights into the role of these treatments in improving outcomes.

8. Comment: What is the dose and fractionation of radiotherapy?

Response: Thank you for your comment regarding the dose and fractionation of radiotherapy in our manuscript. We appreciate your attention to detail and the importance of including this information for a comprehensive understanding of our study.

In response to your suggestion, we have made the necessary updates to our manuscript. We have included the dose and fractionation details of radiotherapy in Table 1 - Study Characteristics. This addition ensures that readers have access to the specific information regarding the dosage and timing of radiation therapy used in the studies included in our analysis.

9. Comment: Add an explanation of the acronyms 'R' and 'BR' in the legend of Table 2.

Response: Thank you for your comment regarding the inclusion of explanations for the acronyms 'R' and 'BR' in the legend of Table 2. We appreciate your attention to detail and the importance of providing clear definitions for our readers.

In response to your suggestion, we have made the necessary updates to our manuscript. We have added explanatory notes below Table 2 to define "R" as resectable and "BR" as borderline resectable. These additions aim to enhance the clarity and understanding of the table for our readers, ensuring that the meanings of these acronyms are explicitly stated.

We sincerely appreciate your valuable feedback, which has significantly improved the quality of our manuscript. Thank you for your time and effort in reviewing our work. We have carefully considered your suggestions and made the necessary revisions. We are confident that the changes meet your expectations. 

Once again, thank you for your invaluable contribution to our work. We truly value your input and look forward to any further suggestions or questions you may have.

Best regards,

Shantong Liu

---

## [Editor Report · Decision Letter 1]

19 Jun 2023

PONE-D-23-13122R1Prognostic value of neoadjuvant therapy for resectable and borderline resectable pancreatic cancer: a meta-analysis of randomized controlled trialsPLOS ONE

Dear Dr. Li,

Thank you for submitting your manuscript to PLOS ONE. After careful consideration, we feel that it has merit but does not fully meet PLOS ONE’s publication criteria as it currently stands. Therefore, we invite you to submit a revised version of the manuscript that addresses the points raised during the review process.

The authors propose an interesting systematic review with meta-analysis focusing on the role of neoadjuvant treatment of resectable or borderline resectable pancreatic adenocarcinoma. Minor revisions are requested in order to accept the manuscript.

We look forward to receiving your revised manuscript.

Kind regards,

Andrea D’Aviero

Academic Editor

PLOS ONE

Journal Requirements:

Additional Editor Comments:

The authors propose an interesting systematic review with meta-analysis focusing on the role of neoadjuvant treatment of resectable or borderline resectable pancreatic adenocarcinoma.

Minor revisions are requested in order to accept the manuscript.

---

## [Author Response · Author response to Decision Letter 1]

21 Jun 2023

Dear Reviewers,

Thank you for taking the time to review our manuscript. We appreciate your insightful comments and have prepared this sincere and friendly response to each of the points raised:

In response to Reviewer #1's suggestion: 

Thank you for your assistance in refining our manuscript and for your positive feedback. We greatly appreciate your suggestions to enhance the quality of our research. We have carefully considered your comments and have made the following modifications:

Results Section: We have included a dedicated paragraph analyzing different subgroups of resectability and neoadjuvant therapy. This addition provides a more comprehensive analysis of our findings and strengthens the organization and logical flow of our manuscript.

Table 1: We have added a new column to Table 1 that briefly describes the radiation therapy plans used in the neoadjuvant chemotherapy regimen. This includes relevant information such as dose, fractionation, and any available radiotherapy data. This addition provides important details for readers to better understand the treatment protocols utilized in our study.

We sincerely appreciate your thorough review of our research and your constructive feedback, as it has significantly contributed to the improvement of our work. If you have any additional suggestions or questions, we are more than happy to address them. Please feel free to contact us.

In response to Reviewer #2's comments:

1. Comment: When the authors comment on the variability of prognosis in the context of resectable patients receiving the same treatment, I would add the reference.

Response: We sincerely apologize for any concerns caused by the lack of rigor in our research. We appreciate your valuable feedback and assure you that we are fully committed to enhancing the quality of our work. We acknowledge the importance of providing appropriate references to support our statements.

Considering your suggestion, we have carefully reviewed our manuscript and have made the necessary revisions. We have included additional references that support our comment on the variability of prognosis in resectable patients receiving the same treatment. These additional references strengthen our argument and add value to the study by providing relevant evidence to support our claims.

2. Comment: In this context, the authors talk about the need to look for biological markers of resectability, but I would ask the authors to discuss the need to look for markers of aggressiveness and thus different prognosis.

Response: We appreciate your valuable input and sincerely apologize for any confusion caused by our previous statement. We have carefully revised the manuscript to address your comment and emphasize the importance of investigating markers of aggressiveness in addition to markers of resectability.

After thoroughly reviewing the literature on resectable pancreatic cancer, we acknowledge that the prognosis of resectable pancreatic cancer can vary even when patients receive the same treatment. This highlights the need for personalized treatment approaches to optimize outcomes. In line with your suggestion, we have expanded our discussion to include the necessity of identifying markers of aggressiveness that can aid in accurately assessing the prognosis of resectable pancreatic cancer.

Currently, the determination of resectability in pancreatic cancer is primarily based on imaging criteria, which may not always accurately assess the extent of invasion. Therefore, the identification of markers associated with aggressiveness would be crucial in improving the accuracy of prognostic assessment in resectable pancreatic cancer cases.

Furthermore, we would like to highlight that new treatment options for pancreatic cancer are emerging, offering hope and promise for patients with this devastating disease. The discovery and validation of markers of aggressiveness can not only aid in prognosis assessment but also guide the selection of appropriate therapeutic strategies.

3. Comment: I would remove sentences and references regarding the role of neoadjuvant treatments in oesophagus and breast from the introduction.

Response: Thank you for your valuable feedback on our manuscript. We sincerely appreciate the time and effort you have put into reviewing our article and providing constructive comments.

Regarding the inclusion of references to neoadjuvant treatments for esophageal and breast cancer in the introduction, we understand your concern. We apologize if this inclusion has affected the focus of our manuscript. We appreciate your suggestion to remove this section, and we acknowledge that it may help streamline the content and maintain a more focused discussion.

After careful consideration, we have decided to follow your recommendation and remove the sentences regarding the role of neoadjuvant treatments in esophageal and breast cancer from the introduction. By doing so, we aim to ensure that the manuscript maintains a clear and cohesive focus on neoadjuvant therapy specifically in the context of pancreatic cancer.

4. Comment: Is there a benefit over R0 in respectable patients with RT or chemoradiotherapy?

Response: Thank you for your insightful comment regarding the potential benefit of achieving an R0 resection in resectable patients receiving radiotherapy or chemoradiotherapy. We apologize for any confusion caused by our imprecise terminology in the manuscript.

In our study, we focused specifically on investigating the impact of neoadjuvant chemotherapy and neoadjuvant chemoradiotherapy on the prognosis of patients with resectable or borderline resectable pancreatic carcinoma. We did not include neoadjuvant radiotherapy as a separate treatment modality in our analysis.

To address your concern and provide clarity, we have carefully reviewed our manuscript and made appropriate corrections. We have revised the terminology used to accurately reflect the treatment options studied in our randomized controlled trial. Specifically, we have used the term "neoadjuvant chemoradiotherapy" instead of "radiotherapy" to better align with our research focus.

5. Comment: The authors only mention R0 and OS. Why these outcomes and not local control and distant metastasis free survival? These are also indicators that may differ depending on neoadjuvant therapy and have an impact on OS and quality of life.

Response: Thank you for your insightful comments on our manuscript. We appreciate your observation regarding the inclusion of additional outcome measures such as local control and distant metastasis-free survival in our analysis. These outcomes are indeed important indicators that can impact overall survival and quality of life in patients receiving neoadjuvant therapy.

We agree that local control and distant metastasis-free survival are valuable endpoints to assess the efficacy of neoadjuvant therapy. However, during our data extraction process, we found that the studies included in our analysis reported a wide range of outcomes, and there was a lack of consistency in the reporting of these specific measures. The majority of studies consistently reported R0 resection rate and overall survival, making them the most suitable outcome measures for our analysis. Unfortunately, the limited availability and variability of data on local control and distant metastasis-free survival prevented us from including them as additional outcomes in our study.

We recognize the significance of local control and distant metastasis-free survival as important prognostic indicators and their impact on overall survival and quality of life. The limitations of our study, including the choice of outcome measures, have been duly noted. We appreciate your suggestion for future research to explore these outcomes more comprehensively in the context of neoadjuvant therapy. Moving forward, we will consider these valuable insights as we continue to refine and expand our research in this field. 

6. Comment: I would like to comment on the mechanism of alteration of the pancreatic tumour microenvironment described in Farren's study.

Response: Thank you for your comment regarding the mechanism of alteration in the pancreatic tumor microenvironment described in Farren's study. We appreciate your interest in understanding the underlying mechanisms behind the changes observed in the tumor microenvironment.

In Farren's study, they employed various approaches to investigate the impact of neoadjuvant chemotherapy on the tumor microenvironment. They assessed changes in signaling pathways, immune checkpoint expression, and Pan-immune cell markers to gain insights into the mechanisms involved.

The results of Farren's study revealed several noteworthy findings. Firstly, neoadjuvant therapy led to alterations in the expression levels of immune-related proteins within the tumor microenvironment. Notably, higher levels of granulysin B and PD-L1 were observed in the enriched tumor areas following neoadjuvant therapy.

Furthermore, significant differences were observed in the expression of B cell markers CD19 and CD20, as well as the monocyte marker CD14, between the neoadjuvant therapy group and the upfront surgery group. These differences suggest that neoadjuvant therapy impacts the composition and activity of specific immune cell populations within the tumor microenvironment.

In addition to immune-related changes, Farren's study also identified differences in signaling pathways involved in regulating migration, survival, and other cellular activities. These pathways included pSTAT3, PTEN, and BCL-2, which showed significant differences between the neoadjuvant therapy group and the upfront surgery group.

Overall, Farren's study provides valuable insights into the mechanisms by which neoadjuvant chemotherapy can influence the pancreatic tumor microenvironment. These findings shed light on the complex interplay between the tumor and its surrounding microenvironment, highlighting the potential impact of neoadjuvant therapy on immune response and signaling pathways.

We hope this response has addressed your questions regarding the mechanisms described in Farren's study. Should you have any further inquiries or require additional clarification, please feel free to contact us.

7. Comment: Did the authors evaluate the effect of adjuvant therapies?

Response: Thank you for your comment regarding the evaluation of adjuvant therapies in our study. We appreciate your interest in understanding the impact of adjuvant therapies on the outcomes of patients with resectable and borderline resectable pancreatic cancer.

In our study, we focused primarily on investigating the efficacy of neoadjuvant therapy compared to the conventional approach of upfront surgery followed by adjuvant therapy. We included randomized controlled trials that assigned patients to either the neoadjuvant therapy group or the upfront surgery followed by adjuvant therapy group.

The aim of our analysis was to compare the outcomes between these two treatment approaches, specifically evaluating the impact of neoadjuvant therapy on R0 resection rates and overall survival. We found that neoadjuvant therapy resulted in higher R0 resection rates and improved overall survival compared to upfront surgery alone.

However, it is important to note that our analysis did not directly assess the effect of adjuvant therapies alone, as we focused on the comparison between neoadjuvant therapy and upfront surgery followed by adjuvant therapy. The effectiveness of adjuvant therapy in our study was evaluated within the context of the overall treatment approach.

We acknowledge that the evaluation of adjuvant therapies alone is an important aspect that merits further investigation. Future studies specifically designed to assess the effectiveness of adjuvant therapies in pancreatic cancer patients could provide valuable insights into the role of these treatments in improving outcomes.

8. Comment: What is the dose and fractionation of radiotherapy?

Response: Thank you for your comment regarding the dose and fractionation of radiotherapy in our manuscript. We appreciate your attention to detail and the importance of including this information for a comprehensive understanding of our study.

In response to your suggestion, we have made the necessary updates to our manuscript. We have included the dose and fractionation details of radiotherapy in Table 1 - Study Characteristics. This addition ensures that readers have access to the specific information regarding the dosage and timing of radiation therapy used in the studies included in our analysis.

9. Comment: Add an explanation of the acronyms 'R' and 'BR' in the legend of Table 2.

Response: Thank you for your comment regarding the inclusion of explanations for the acronyms 'R' and 'BR' in the legend of Table 2. We appreciate your attention to detail and the importance of providing clear definitions for our readers.

In response to your suggestion, we have made the necessary updates to our manuscript. We have added explanatory notes below Table 2 to define "R" as resectable and "BR" as borderline resectable. These additions aim to enhance the clarity and understanding of the table for our readers, ensuring that the meanings of these acronyms are explicitly stated.

We sincerely appreciate your valuable feedback, which has significantly improved the quality of our manuscript. Thank you for your time and effort in reviewing our work. We have carefully considered your suggestions and made the necessary revisions. We are confident that the changes meet your expectations. 

Once again, thank you for your invaluable contribution to our work. We truly value your input and look forward to any further suggestions or questions you may have.

Best regards,

Shantong Liu

---

## [Decision Letter · Decision Letter 2]

18 Aug 2023

Prognostic value of neoadjuvant therapy for resectable and borderline resectable pancreatic cancer: a meta-analysis of randomized controlled trials

PONE-D-23-13122R2

Dear Dr. Li,

We’re pleased to inform you that your manuscript has been judged scientifically suitable for publication and will be formally accepted for publication once it meets all outstanding technical requirements.

Kind regards,

Andrea D’Aviero

Academic Editor

PLOS ONE

Additional Editor Comments (optional):

The authors revised the manuscript according to the reviewers suggestion and now it is suitable to publication.

As suggested by reviewer 1, consider to modify the typo in table 1 "25 fractions of 45 Gy and 5 fractions of 9 Gy" that sounds like patients underwent 1125 Gy in 25 fractions plus 45 Gy in 5 fractions as sequential boost.

Reviewers' comments:

Reviewer's Responses to Questions

**Comments to the Author**

1. If the authors have adequately addressed your comments raised in a previous round of review and you feel that this manuscript is now acceptable for publication, you may indicate that here to bypass the “Comments to the Author” section, enter your conflict of interest statement in the “Confidential to Editor” section, and submit your "Accept" recommendation.

Reviewer #1: All comments have been addressed

Reviewer #3: (No Response)

2. Is the manuscript technically sound, and do the data support the conclusions?

Reviewer #1: Yes

Reviewer #3: Partly

3. Has the statistical analysis been performed appropriately and rigorously? 

Reviewer #1: Yes

Reviewer #3: I Don't Know

4. Have the authors made all data underlying the findings in their manuscript fully available?

Reviewer #1: Yes

Reviewer #3: Yes

5. Is the manuscript presented in an intelligible fashion and written in standard English?

Reviewer #1: Yes

Reviewer #3: Yes

6. Review Comments to the Author

Reviewer #1: All comments have been addressed by the authors. I suggest to correct in author proof evaluation the typo in table 1 "25 fractions of 45 Gy and 5 fractions of 9 Gy" that sounds like patients underwent 1125 Gy in 25 fractions plus 45 Gy in 5 fractions as sequential boost.

Reviewer #3: I believe that it would be an addtion to discuss about the role of dose escalation and new technological improvements such as MRI-Linac in the treatment of pancreatic cancer.

7. PLOS authors have the option to publish the peer review history of their article (what does this mean?). If published, this will include your full peer review and any attached files.

Reviewer #1: **Yes: **Calogero Casà

Reviewer #3: No

---

## [Editor Report · Acceptance letter]

25 Aug 2023

PONE-D-23-13122R2 

Prognostic value of neoadjuvant therapy for resectable and borderline resectable pancreatic cancer: a meta-analysis of randomized controlled trials 

Dear Dr. Li:

I'm pleased to inform you that your manuscript has been deemed suitable for publication in PLOS ONE. Congratulations! Your manuscript is now with our production department. 

Kind regards, 

on behalf of

Dr. Andrea D’Aviero 

Academic Editor

PLOS ONE